# Probability Turns Material: The Boltzmann Equation

**DOI:** 10.3390/e26020171

**Published:** 2024-02-17

**Authors:** Lamberto Rondoni, Vincenzo Di Florio

**Affiliations:** 1Dipartimento di Scienze Matematiche, Politecnico di Torino, Corso Duca Degli Abruzzi 24, 10129 Turin, Italy; 2INFN, Sezione di Torino, Via Pietro Giuria 1, 10125 Turin, Italy; 3Istituto Italiano di Tecnologia (IIT), Via Melen 83, B Block, 16163 Genoa, Italy

**Keywords:** dynamical systems, probability, observables

## Abstract

We review, under a modern light, the conditions that render the Boltzmann equation applicable. These are conditions that permit probability to behave like mass, thereby possessing clear and concrete content, whereas generally, this is not the case. Because science and technology are increasingly interested in small systems that violate the conditions of the Boltzmann equation, probability appears to be the only mathematical tool suitable for treating them. Therefore, Boltzmann’s teachings remain relevant, and the present analysis provides a critical perspective useful for accurately interpreting the results of current applications of statistical mechanics.

## 1. Introduction

Statistical mechanics arose to interpret thermodynamic results from an atomistic perspective. That is an objectively difficult task because thermodynamics is deterministic. It views matter as a continuum characterized by a reduced number of variables, and it is irreversible in time. Partial differential equations are the suitable mathematical tool for such a description of physical objects. On the contrary, the microscopic dynamics of atoms are considered deterministic when dealing with fluids in non-extreme conditions. However, they are time-reversal invariant, with a few notable exceptions, such as Prigogine [1]. Being discrete rather than continuous, these dynamics are represented by systems of many ordinary differential equations. The difficulty, then, does not lie in doubts about the atomistic hypothesis, which have been dispelled long ago [2], when the Brownian motion was explained [3]. The difficulty concerns the above very different mathematical descriptions that have been developed for macroscopic and microscopic dynamics. Indeed, they are so different that they could even look incompatible.

With statistical physics, the gap has been filled adding statistical notions to the purely mechanical ones, and profiting from the mind-boggling number of degrees of freedom that must be integrated out to go from atoms to continuum [4]. This second aspect was initially overlooked by Boltzmann’s contemporaries, who criticized him on the grounds of the laws of mechanics [5]. As a matter of fact, certain sentences in Boltzmann’s fundamental paper [6] prompted criticism because they lack reference to probabilistic arguments and seem to imply a direct derivation of irreversible relaxation to equilibrium from the purely reversible dynamics of atoms:

*“It has thus been rigorously proved that, whatever may be the initial distribution of kinetic energy, in the course of a very long time it must always necessarily approach the one found by Maxwell. The procedure used so far is of course nothing more than a mathematical artifice employed in order to give a rigorous proof of a theorem whose exact proof has not previously been found. It gains meaning by its applicability to the theory of polyatomic gas molecules”*.

Whether this interpretation of Boltzmann’s intention is correct or not, Loschmidt and Zermelo legitimately, and correctly, pointed out that mechanics by itself does not justify the result. Boltzmann’s reply, then, set things straight once and for all. Although, to date, the argument can be considered quite subtle. In a sense, Boltzmann agreed with the criticism, without admitting any error in his original point of view, but simply elaborating on it. He invoked a probabilistic argument, which does not belong to mechanics, but is powerfully unavoidable: given the number of molecules that make a macroscopic object, it is unreasonable not to take seriously the probabilistic predictions. In his own words, this is Boltzmann’s answer to Zermelo:

*“The applicability of probability theory to a particular case cannot of course be proved rigorously. If, out of 100,000 objects of a certain kind, about 100 are annually destroyed by fire, then we cannot be sure that this will happen next year. On the contrary, if the same conditions could be maintained for $10^{10^{10}}$ years, then during this time it would often happen that all 100,000 objects would burn up on the same day; and likewise there will be entire years during which not a single object is damaged. Despite this, every insurance company relies on probability theory. It is even more valid, on account of the huge number of molecules in a cubic millimetre, to adopt the assumption (which cannot be proved mathematically for any particular case) that when two gases of different kinds or at different temperatures are brought in contact, each molecule will have all the possible different states corresponding to the laws of probability and determined by the average values at the place in question, during a long period of time. These probability arguments cannot replace a direct analysis of the motion of each molecule; yet if one starts with a variety of initial conditions, all corresponding to the same average values (and therefore equivalent from the viewpoint of observation), one is entitled to expect that the results of both methods will agree, aside from some individual exceptions which will be even rarer than in the above example of 100,000 objects all burning on the same day. The assumption that these rare cases are not observed in nature is not strictly provable (nor is the entire mechanical picture itself) but in view of what has been said it is so natural and obvious, and so much in agreement with all experience with probabilities, from the method of least squares to the dice game, that any doubt on this point certainly cannot put in question the validity of the theory when it is otherwise so useful. It is completely incomprehensible to me how anyone can see a refutation of the applicability of probability theory in the fact that some other argument shows that exceptions must occur now and then over a period of eons of time; for probability theory itself teaches just the same thing”*.

(We would like to stress one aspect of this reply that is also quite important, although not directly related to the debate between Boltzmann and Zermelo, hence usually not considered. Boltzmann clearly indicates that a mathematical explanation of a physical phenomenon cannot be taken as applicable without restrictions. Indeed, any mathematical model is just that: it cannot replace reality in all its aspects. It suffices to keep this in mind to realize paradoxes may simply lie in the incorrect application of a theory to situations exiting its range of validity. The problem may be not lack of mathematical rigour, but of physical relevance.) One should note that, strictly speaking, Boltzmann reasonings apply to a special class of macroscopic objects, the rarefied gases. Nevertheless, over the years, the Boltzmann equation has been extended to an incredibly wide range of systems and phenomena. First, within the realm of physics, it has been extended to classical systems subjected to external forces, as well as to quantum and relativistic mechanics, with applications, e.g., in solid state physics and astrophysics, cf., e.g., [7,8]. Successively, applications have been produced in fields such as social sciences, economics, and traffic studies [9,10,11,12,13], just to mention a few examples. In mathematics, the study of the Boltzmann equation has also produced fundamental results, of general interest. For instance, the proof of the existence of solutions [14]. More recently, Cercignani’s conjecture [15], that has been influential in understanding dissipation from a microscopic point of view, was proven to be substantially correct [16].

Even through all that, one comes to realize that probability, the primary tool of statistical physics, is crucial for accurately handling quantities related to discrete lumps of matter composed of numerous microscopic constituents, especially on the scale where they seem to exhibit continuous behavior. But, thanks to this tool, statistical physics may in principle be used even beyond of the realm of thermodynamics if is it true, as it is, that even insurance companies rely on it. In fact, statistical physics has turned its attention more and more to the study of small systems, which are not guaranteed to be in Local Thermodynamic Equilibrium (LTE), and require thermodynamic quantities to be interpreted in a different or wider sense than in thermodynamics. Indeed, LTE requires, at least, that systems should possess a microscopic, mesoscopic and macroscopic scale in space and time w.r.t the observer. This condition cannot be ensured for all systems to which statistical physics is applied. This can lead, for instance, to finite size effects and fluctuations entailing blurred boundaries between different states of aggregation matter. This gives probabilities an even wider role in the description of small systems than in the statistical mechanics of macroscopic objects. Often, probability is the only sensible thing to compute. The trouble is that probability is immaterial.

In this paper, we analyze various notions of probability, highlighting this fact, and then deomnstrate how it may be seen turning material within the framework of the Boltzmann equation. This may be useful in correctly interpreting a large fraction of present day statistical mechanics results.

## 2. Probability: Obscure and Practical

How can probability be useful? Karl Popper maintained that probabilistic results cannot be falsified, thus contradicting his concept of science: *For although probability statements play such a vitally important role in empirical science, they turn out to be in principle impervious to strict falsification”* [17]. Further:


*“In whatever way we may define the concept of probability, or whatever axiomatic formulations we choose: so long as the binomial formula is derivable within the system, probability statements will not be falsifiable. Probability hypotheses do not rule out anything observable; probability estimates cannot contradict, or be contradicted by, a basic statement; nor can they be contradicted by a conjunction of any finite number of basic statements; and accordingly not by any finite number of observations either”*
[17].

Of course, Popper’s view of science can be criticized, but there is a fact: suppose meteorologists have computed that the probability of rain tomorrow is 0.999. We go out carrying an umbrella, and we find the Sun is shining. Have we refuted the meteorologists’ prediction? Of course not. Consequently, if it rains it also does not prove anything about the mentioned calculations. So, why is probability interesting?

In the mathematical sciences, probability is a rather recent subject. In the XVI century, it was conceived to improve chances of winning in gambling. However, it rapidly came to constitute the basis of statistical and quantum mechanics, as well as of chaos theory and of the sciences of “complexity”. It is dominant in finance, in the interpretation of medical tests, in epidemiology, weather forecasts, etc. On the other hand, can we grasp the meaning of a statement like:“there is a probability of $10^{-40}$ that a nuclear power plant blows up”? When and how will we experience a number like that, so that we develop an opinion about its significance? What do scientists say?

Joh Archibald Wheeler, who would constantly make use of probabilities in his scientific activity, stated that *“Probability, like time, is a concept invented by humans, and humans have to bear the responsibility for the obscurities that attend it. Obscurities there are whether we consider probability defined as frequency or defined a la Bayes”* [18].

The great probabilist, Bruno de Finetti, in the preface of his treatise on probability theory said,


*My thesis, paradoxically, and a little provocatively, but nonetheless genuinely, is simply this:*



*PROBABILITY DOES NOT EXIST*



*The abandonment of superstitious beliefs about the existence of Phlogiston, the Cosmic Ether, Absolute Space and Time, …, or Fairies and Witches, was an essential step along the road to scientific thinking. Probability, too, if regarded as something endowed with some kind of objective existence, is no less a misleading misconception, an illusory attempt to exteriorize or materialize our true probabilistic beliefs*
[19].

Clearly, profound familiarity with the subject does not eliminate obscurities, and probability appears *immaterial*. Simple considerations may help us understand these statements.

The classical notion of probability gives an objective rule to attribute a probability to some expected event *E*. For **finitely**
**many** possible events, *N* say, the classical probability is objectively defined by counting as
(1)$$\mathrm{Prob}\left(E\right)=\frac{N(E)}{N}$$
where N(E) is number of possible outcomes yielding *E*. This means that the single events are declared equally probable. Coin tossing is the usual example of this kind: the single events are *heads* and *tails*; therefore, each of them is attributed a probability of 1/2. It is a simple and appealing notion of probability, that looks particularly useful to deal with situations that totally defeat our understanding. However, it is obviously not suitable e.g., for tossing loaded coins or dice; or for YES or NO answers to “Is it going to rain?” Moreover, paradoxes arise if the set of possible events cannot be counted, particularly for continuous sets of events. Think, for instance, of the Bertrand Paradox: there are three different, apparently equally legitimate, probabilities for the event *L*, which represents the probability that a chord of a circle of unit radius is longer than $\sqrt{3}$, the length of the side of the inscribed equilateral triangle:given two points on the circumference, consider the chord joining them, and take one of the two points as the vertex of one inscribed equilateral triangle. If the chord lies outside the triangle, it is shorter than $\sqrt{3}$; if it intersects the triangle, it is longer than or equal to $\sqrt{3}$. Now, the vertices of the triangle delimit three arcs of equal length, and only one of them corresponds to a chord longer than $\sqrt{3}$. Therefore, the probability of *L* is 1/3.consider a radius of the circle, and choose a point on it. There is a chord perpendicular to this radius, and its length is larger than $\sqrt{3}$ if the point is closer to the center of the circle than to the circumference. Therefore, the probability of *L* is 1/2.take a point anywhere within the circle and construct a chord with this point as its midpoint. The chord is longer than $\sqrt{3}$ if the point lies within a concentric disk of radius 1/2 the radius of the larger circle. The area of the smaller circle is 1/4 of the full disk; therefore, the probability of *L* is 1/4.

Despite the numerous *solutions* that have been devised to solve this conundrum, see, e.g., Ref. [20], it remains that they all refer to some kind of superior principle that can only be subjective. Hence, they cannot be taken as objectively valid, at least not in the sense of the mass of a stone. One may then adopt a subjective notion of probability at the start. This amounts to saying that probability should be assigned by the one interested in it, defining
Prob(E)=maximum someone consistently puts at stake to win 1 in an unbiased game
There is an immediate advantage with respect to the classical notion of probability: it is always possible to unambiguously associate a probability with an event, even in the question of whether or not I am a robot. But there are difficulties as well. First of all, different people may have different views on the event *E*, and that may make the scientific activity cumbersome.

In some cases, one may think that probabilities are objectively imposed by the sequence of events. One then may rely on a different kinds of counting events, thinking that at each time only a finite number of events has occurred, but assuming that extrapolation to infinitely many events is possible. This constitutes the frequentist approach. Let $N_{t}$ be the number of observations, or experiments repeated in time up to time *t*. Then,
$$\mathrm{Prob}\left(E\right)=\lim_{t\to \infty }\frac{N(E;t)}{N_{t}}$$
where N(E;t) is the number of outcomes regarding *E* in the time *t*. This notion of probability is very suggestive e.g., in dealing with situations that are often repeated in time or space, such as weather forecasts or car accidents. However, many events happen only once or a few times in a lifetime, and can be dramatically positive or negative, like nuclear power plants accidents or becoming Captain Regent of the San Marino Republic. Even harder problems are posed by events that happen frequently, but not enough for us to extract a reliable predictive rule, e.g., earthquakes or pandemics.

From these considerations, one might ask why bother at all to use probabilities. The fact is that the above has not stopped people from referring to probabilities. This is understandable if one realizes that probability, like any other mathematical tool, yields a model, one description of one aspect of reality, and does not intend to represent the whole reality. However, as such, it can change its meaning depending on the problem at hand; and it can do so without causing any inconvenience, enabling us to better understand the problems at hand. Indeed, James Clerk Maxwell argued that


*They say that Understanding ought to work by the rules of right reason. These rules are, or ought to be, contained in Logic; but the actual science of Logic is conversant at present only with things either certain, impossible, or entirely doubtful, none of which (fortunately) we have to reason on. Therefore the true Logic for this world is the Calculus of Probabilities, which takes account of the magnitude of the probability (which is, or which ought to be in a reasonable man’s mind). This branch of Math., which is generally thought to favour gambling, dicing, and wagering, and therefore highly immoral, is the only ‘Mathematics for Practical Men’, as we ought to be*
[21].

How can one be practical? The approach that has proven successful is refraining from adopting a specific interpretation of probability, in general, and to develop a mathematical theory within which unambiguous conclusions can be reached, by means of suitable calculations. In doing so, one must also bear in mind that results will refer to a rational framework, not to reality, and connection with reality will have to be assessed case by case. The mathematical framework suitable to this task is that of axiomatic probability, briefly summarized as follows:There is a set Ω called sample space, which contains all the “elementary” events ω;there is a collection of subsets of Ω, F(Ω) that has the structure of a σ− algebra of subsets E⊂Ω, representing all (elementary and combined) events to which a probability is to be assigned;for all E∈F, P(E)≥0, P(∅)=0, and P(Ω)=1, where *∅* is the empty set;for $E_{1}\cap E_{2}=\emptyset$ one has $P\left(E_{1}\cup E_{2}\right)=P\left(E_{1}\right)+P\left(E_{2}\right)$;given an infnite sequence of disjoint events, ${\left\{E_{i}\right\}}_{i=1}^{\infty }$ with $E_{i}\cap E_{j}=\emptyset$ if i≠j, one has
(2)$$P\left(\;\overset{\infty }{\underset{i=1}{⋃}}E_{i}\right)=\sum_{i=1}^{\infty }P\left(E_{i}\right)$$

Given these axioms, the rest follows, provided one has a suitable method of assigning probabilities to the elementary events. However, this method does not necessarily need to be a general rule endowed with a specific interpretation.

When do we need probabilities? When events (apparently) follow no rule, like the outcomes of coin tossing, or the trajectories of pollen grains in water, that are called random phenomena. Singularly, such phenomena follow no rule, but it may happen that their collective large-scale behaviour is described by (possibly evolving) probabilities that follow precise rules, which are called stochastic processes. Deterministic processes, on the other hand, are singularly subjected to strict rules, such as those of classical mechanics, in which the future is totally determined by the initial conditions. However, because such evolutions require exact knowledge of the initial conditions, something that is never achievable in practice, probabilities play a fundamental role also in the case of deterministic evolutions, especially those that are known as chaotic. As a matter of fact, determinism and randomness are but points of view, more or less convenient, depending on many factors. One cannot empirically decide the “true” nature of a given phenomenon. One can only try to make our descriptions effective: i.e., satisfactory for our purposes, and we are satisfied when we can make reliable predictions.

Let us investigate how such an elusive and immaterial notion as probability became concrete and, in fact, “material” thanks to Boltzmann’s work and the Boltzmann equation. This may then serve as a guide to the use of probabilities in realms in which they often are the only suitable mathematical tool, but still bear the consequences of being “obscure” and “immaterial”.

## 3. Mass vs. Probability

Consider *N* particles, each with *d* degrees of freedom, which obey the equations of motion $\dot{x}=G\left(x\right)$, on their phase space M, which is *n*-dimensional, with n=2dN. Let $\Phi^{t}$ represent the operator, which yields the solution of the equation of motion at time *t*, $\Phi^{t}x$, if *x* is the initial condition. The functions of phase, O:M⟶R, are considered microscopic properties of such particles. Then, time averages
(3)$$\bar{O}\left(x\right)=\lim_{t\to \infty }\frac{1}{t}\int_{0}^{t}O\left(\Phi^{s}x\right)ds$$
are postulated to represent macroscopically observable quantities. The idea is that a measurement tool and the macroscopic object of interest interact for a macroscopic time, which is very long compared to microscopic times, and the result is represented by the time average of the quantity of interest. Assuming that one is treating objects amenable to thermodynamic analysis, which are described by a handful of observables, the t→∞ limit results in a very good mathematical idealization of the measurement time, since that may last $10^{12}$ or more microscopic units. Moreover, the dependence on the initial condition *x* of a trajectory, $\gamma \left(x\right)={\left\{\Phi^{t}x\right\}}_{t\in R}$, representing the microscopic history of a given (single) object of interest, is irrelevant. (This is not necessarily true in small systems, in which local thermodynamic equilibrium is not verified; hence, thermodynamic relations may be inapplicable.) This suggests that the time average (Equation 3) may be obtained by replacing the impossible-to-compute phase space trajectory with a phase space probability distribution μ that weighs the different regions of M proportionally to the frequency with which γ(x) visits them:(4)$$\bar{O}\left(x\right)=\int_{M}Od\mu =E_{\mu }\left[O\right]\;.$$
Here, the right hand side is called the phase space average. Because the left hand side of Equation (Equation 4) depends on *x*, while the right hand side does not, one must specify that the equality is not expected to hold for all x∈M. At most, one can ask that it holds for *μ-almost*
*every* x∈M, i.e., as long as its possible violations are confined within a set E∈M of vanishing probability: $$\mu \left(E\right)=\int_{E}d\mu =0,$$
which by definition does not affect the integral in Equation (Equation 4).

The investigation of the validity of Equation (Equation 4) has led to the development of ergodic theory, concerning the properties of dynamical systems and their invariant measures. In its standard formulation, this mathematical theory states that there are four equivalent properties called *ergodicity*, including the validity of Equation (Equation 4). Letting (M,μ,Φ) be a dynamical system with μ, an invariant measure under the dynamics Φ defined on the phase space M, the following properties are equivalent:**(E1)** For every integrable function O:M↦R, time average and phase space average coincide for μ -almost every *x*, which means for all x∈M apart from a set *E* of vanishing measure, i.e., such that μ(E)=0:
(5)$$\bar{O}\left(x\right)=E_{\mu }\left(O\right)\;,\;\mu -a.e.\;x\in M$$**(E2)** For every measurable A⊂M and for μ -a.e. x∈M, the fraction of time a trajectory spends in *A* is given by:
(6)$$\tau_{A}\left(x\right)=\mu \left(A\right)\;,\;\mu -a.e.\;x\in M$$
where the fraction of time is defined by
(7)$$\tau_{A}\left(x\right)=\lim_{t\to \infty }\frac{1}{t}\int_{0}^{t}\chi_{A}\left(\Phi^{s}x\right)ds$$**(E3)** There are no non-trivial integrals of motion. In other words, let O be a function of phase, such that $O\left(\Phi^{t}x\right)=O\left(x\right)$ for all *t* and for μ -a.e. x∈M, and let O be integrable. Then,
(8)O(x)=constant,μ−a.e.x∈M**(E4)** The dynamical system is metrically indecomposable. In other words, let *A* be an invariant measurable set, i.e., $\Phi^{t}A=A$ for all *t*. Then, either μ(A)=1 or μ(A)=0. When that is the case, the expression M=A∪(M∖A) is called metrically trivial decomposition of M.

It is easy to see that ergodicity may be realized in simple dynamical systems whose phase space trajectories γ(x) move parallel to each other and with the same phase space “velocity”, i.e., same G(x). Take for instance a two-dimensional torus as M, and let trajectories be oriented so that they do not close on a periodic orbit. (Cutting the torus to obtain a square, with periodic boundary conditions, it suffices that the components of *G* be not rational with respect to each other.) This is but an example of an ergodic system whose trajectories remain close to each other in time, while they explore the whole phase space, thus satisfying E2. Now, imagine that probability belongs to phase space sets like mass belongs to chunks of matter moving in space. In other words, let us mimic the conservation of mass in real space with the conservation of probability in phase space, although the difference between the two is huge, as is shown even by the dimensions of real space, which is 3, and of M, which is $O(2\times 3\times 10^{23})$ for spherical atoms. That means that probabilities may evolve, obeying
(9)$$\mu_{t}\left(E\right)=\mu_{0}\left(\Phi^{-t}E\right)$$
and, if the evolving probability has a density, it satisfies the (generalized) Liouville equation
(10)$$\frac{\partial f}{\partial t}=-{\mathrm{div}}_{\Gamma }\left(fG\right)=-G\cdot \nabla_{\Gamma }f-f\;{\mathrm{div}}_{\Gamma }G$$
If the equilibrium distribution is the microcanonical ensemble, which, for an ideal gas, is the uniform distribution of microstates with specific energy, and if the initial distribution is not uniform, the ergodic property does not lead in time to the equilibrium distribution. In our example of parallel trajectories, a lump of probability continues to go round and round in M, never spreading to become uniform. This will be revealed by the phase space averages of observables that will correspondingly fluctuate in time, as the support of this initial probability moves within M. If one pushes further the analogy between probability and mass, taking the spread of probability in phase space as representing the spread of the molecules of a gas in their container, ergodicity does not suffice to describe the *irreversible* relaxation of macroscopic states to their equilibrium. Therefore, a stronger property, that implies ergodicity, has been considered to describe the most common irreversible phenomenon, see, e.g., Ref. [22]. This condition is called *mixing*, and is defined by two equivalent properties:**(M1)** For every pair of measurable sets in the phase space, A,B⊂M, the following holds:
(11)$$\lim_{t\to \infty }\mu \left(\Phi^{-t}A\cap B\right)=\mu \left(A\right)\mu \left(B\right)\;,\;\mathrm{or}\;\;\;\lim_{t\to \infty }\mu \left(A\cap \Phi^{t}B\right)=\mu \left(A\right)\mu \left(B\right)$$**(M2)** for every pair of observables, e.g., $O,P\in L_{2}\left(M,\mu \right)$, the following holds:
(12)$$\lim_{t\to \infty }\int_{M}\left(O\circ \Phi^{t}\right)Pd\mu =\int_{M}Od\mu \int_{M}Pd\mu \;,$$
i.e., $\lim_{t\to \infty }E_{\mu }\left[(O\circ \Phi^{t})P\right]=E_{\mu }\left(O\right)E_{\mu }\left(P\right)$ 

The meaning of M1 is that, in time $\Phi^{t},B$ will occupy a fraction of *A* equal to its measure in the full M. As *A* can be taken in any region of phase space, in particular within the support of an invariant measure under consideration, this means that an initial ensemble supported in *B* will reach in time and will stay in any corner of the support of μ. When the microcanonical ensemble is the relevant invariant measure, this means that the probability “contained” in *B* will eventually be uniformly distributed over the whole M. This can be seen observing that the invariance of μ implies $\mu \left(A\cap \Phi^{t}B\right)=\mu \left(\Phi^{-t}\left(A\cap \Phi^{t}B\right)\right)=\mu \left(\Phi^{-t}A\cap B\right)$, and $\mu \left(A\right)\mu \left(\Phi^{t}B\right)=\mu \left(A\right)\mu \left(B\right)$. Then, for sufficiently long times, one may approximately write
(13)$$\frac{\mu \left(A\cap \Phi^{t}B\right)}{\mu (A)}\approx \mu \left(B\right)\;,\;\mathrm{provided}\;\;\mu \left(A\right)\neq 0$$
which becomes exact in the t→∞ limit.

Property M2, in particular, expresses the decay of correlations of microscopic events in phase space. When the correlation function of the functions O and P, defined by
(14)$$G\left(t\right)=E_{\mu }\left[(O\circ \Phi^{t})P\right]-E_{\mu }\left(O\right)E_{\mu }\left(P\right)$$
does not vanish, the measurements of O and P are, in a sense, not independent. The decay in time of such correlation indicates a sort of irreversible loss of memory about the initial state. Indeed, a simple calculation appears to confirm that relaxation to an equilibrium state, for a Hamiltonian particle system, is achieved under mixing [23]:(15)$$\begin{array}{ccc} & & \;E_{\mu_{t}}\left(O\right)=\int_{M}O\rho_{t}dx=\int_{M}O\left(\rho_{0}\circ \Phi^{-t}\right)dx\\ & & \;=\int_{M}\left(O\circ \Phi^{t}\right)\rho_{0}dx\;\overset{t\to \infty }{⟶}\;\int_{M}Odx\int_{M}\rho_{0}dx=E_{dx}\left(O\right). \end{array}$$

Here, the dynamics is assumed to be phase space volume-preserving, so that the Jacobian determinant denoted by $J^{-t}$ equals 1. Then, $\rho_{0}=$ constant is the invariant probability density (its integral equals 1), and mixing has been applied, considering the probability density as a single observable. The result is that the initial value of the observable irreversibly converges to its microcanonical value. A mixing counterpart to the uniform translation on the two-dimensional torus is the so-called Arnold’s CAT map, defined by
(16)$$\left(\begin{array}{c} x^{\prime }\\ y^{\prime } \end{array}\right)=\left(\begin{array}{ccc} 1 & & 1\\ 1 & & 2 \end{array}\right)\left(\begin{array}{c} x\\ y \end{array}\right)\;\;\mathrm{mod}1\;.$$

Despite the apparent formal consistency, numerous difficulties arise with these phase space calculations. Although the same rules applying to mass have been applied to probability, the second obviously remains an *immaterial* abstract concept. For instance, mixing also holds for systems made of a single particle confined within convex walls [24], which surely cannot expand to fill its container, while the probability does. One conceptual difficulty with Equation (Equation 15) is that the probability density $\rho_{t}$ is treated like a microscopic physical property of a given object, which it is not. It represents a probability, that is a completely different object. One microscopic quantity, the kinetic energy say, is an objective quantity defined for a single concrete object of interest, which is represented by a single geometric point in the phase space. On the other hand, a probability may either refer to collections of systems, or sequences of experiments, each represented by a single phase in the phase space, or it can be subjectively introduced. In all cases, a distribution of phases in phase space is obtained, but the degree of crowding of phase points in different regions has no bearing on the evolution of a given single system. For example, beyond the cases of systems with a single particle, numerous large systems (high probability) could exist within a small region of phase space, each corresponding to uniform mass density. In such instances, the probability would evolve while the mass of the systems of interest remains unchanged. The evolution of phase space probabilities is not directly related to what pertains to thermodynamic quantities. That is solely determined by the equations of motion and the corresponding initial condition. Being associated with a higher or lower amount of companion phases does not modify its fate.

The question arises: under which conditions does probability turn *material*? Indeed, acknowledging, as Maxwell implied, that we have no better tool than probability does not imply that it can be used uncritically. We get a glimpse of that in Poincaré’s considerations on the laws that govern the large number of molecules constituting a gas:


*“It seems at first that the orderless collisions of this innumerable dust can only engender an inextricable chaos before which the analyst must retire. But the law of great numbers, that supreme law of chance, comes to our assistance. In face of a semi-disorder we should be forced to despair, but in extreme disorder this statistical law re-establishes a kind of average or mean order in which the mind can find itself again”*
[25].

Understanding these facts definitely helps in uncharted territories. Great numbers and extreme disorder seem to be essential, but often they are not present.

## 4. Hamiltonian Particle Systems and Ensembles

Because a priori it is not possible to decide which microscopic dynamics are better suited to the task of describing macroscopic systems, one almost universally begins with classical Hamiltonian mechanics, proven extremely successful in describing an incredible variety of macroscopic phenomena. This was Laplace’s opinion:


*“We may regard the present state of the universe as the effect of its past and the cause of its future. An intellect which at a certain moment would know all forces that set nature in motion, and all positions of all items of which nature is composed, if this intellect were also vast enough to submit these data to analysis, it would embrace in a single formula the movements of the greatest bodies of the universe and those of the tiniest atom; for such an intellect nothing would be uncertain and the future just like the past would be present before its eyes”.*


(Note, however, that Laplace made that statement in an essay on probabilities. Indeed, he considered such an intellect out of our reach. This was not the manifesto of reductionism, as it is often portrayed, but the vindication of probabilities [26]). This was a very bold statement, considering the fact that experience was limited to large, observable objects, which typically come in small numbers, like planets and comets. In principle, there is no relation with the exceedingly large assemblies of invisibly tiny molecules that constitute a macroscopic body. This may be a cause of concern; hence, it must be subjected to testing whenever new phenomena are considered. The outcome is that statistical mechanics has greatly advanced by relying on Hamiltonian dynamics.

Consider a system of *N* particles, each having *d* degrees of freedom, characterized by a Hamiltonian H=H(Γ), where Γ=(q,p)∈M, and by the following equations of motion:(17)$$\dot{q}=\frac{\partial H}{\partial p}\;,\;\;\dot{p}=-\frac{\partial H}{\partial q}$$
As already mentioned, the connection with physical measurements begins from the observation that they take a positive time, and from the assumption that they yield the time average of one phase function. If the measurement takes a time τ, and it starts when the microstate is Γ, the result of the measurement is postulated to be
(18)$${\bar{O}}_{0,\tau }\left(\Gamma \right)=\frac{1}{\tau }O_{0,\tau }=\frac{1}{\tau }\int_{0}^{\tau }O\left(\Phi^{t}\Gamma \right)\;dt$$
where by the symbol $O_{s,t}$ we mean the time integral of O from time *s* to time *t*, and the bar denotes time averaging.

If one accepts this picture, it follows that measurements depend on τ and are therefore subjective because different observers may choose different initial times and different τ’s. They also depend on Γ, hence their result is a random variable because Γ is unknown. However, this contradicts Thermodynamics, which is an objective and deterministic theory, universally confirmed by experimental tests. One possibility to accord Equation (Equation 18) with experience is to assume that τ is very large, virtually infinitely larger than the microscopic scales concerning the evolution of Γ, and that over such a long time, O has explored many times—hence with proper frequencies—its range of values so that the initial condition Γ is irrelevant. The same would hold if, in the time τ, generic phase space trajectories, with negligible exceptions, thoroughly explore the phase space, which is property E2 of Section 3. However, while E2 is sufficient for the result, it is not necessary, and typically impossible. One way or another, we obtain
(19)$$\frac{1}{\tau }\int_{0}^{\tau }O\left(\Phi^{t}\Gamma \right)\;dt\approx \bar{O}\left(\Gamma \right)=\lim_{\tau \to \infty }\frac{1}{\tau }\int_{0}^{\tau }O\left(\Phi^{t}\Gamma \right)\;dt\approx o\in I\;\;R$$
where mathematically the first approximate equality is due to the fact that τ should be very large, although it cannot be infinite, and the second to the fact that the range of O may not be perfectly explored. As long as the approximations fall below the scale of thermodynamic interest, equality can thus be used, and *o* represents the result of a measurement. This picture is justified by Fermi as follows:


*“Studying the thermodynamical state of a homogeneous fluid of given volume at given temperature […] we observe that there is an infinite number of states of molecular motion that correspond to it. With increasing time, the system exists successively in all the dynamical states that correspond to the given thermodynamical state. From this point of view we may say that a thermodynamical state is the ensemble of all the dynamical states through which, as a result of the molecular motion, the system is rapidly passing”*
[27].

Callen adds that, if the system passes sufficiently rapidly through representative atomic states, a measurement effectively averages over all them, and the system is in equilibrium [28]. If, on the other hand, the transitions from atomic state to atomic state is ineffective (e.g., the system is trapped in a small subset of states, or the transition rate is too slow) a macroscopic measurement does not yield a proper average over all possible atomic states, and the system is not in equilibrium. A different description must be adopted in such situations, which are in fact well known; think, e.g., of metal alloys subjected to thermal treatments. However, when the microscopic values of O are indeed sufficiently rapidly explored, compared to macroscopic observation times, a single system of interest reveals itself through O as the average over the *ensemble* of such possibilities. To the observer, the system revealed by O *appears* like the average over the ensemble of its microscopic phases, and the result of the measurement equals an average with respect to a phase space probability distribution, called *ensemble*. However, this implicitly requires the macroscopic state to be stationary; if it shifts during the measurement, the observable cannot explore its range with the frequencies corresponding to that state, and different initial microstates may lead to different observable values. In fact, this is the case of, e.g., ageing systems.

Given a dynamical system like Equation (Equation 17), it is not obvious that Equation (Equation 19) holds, and even if it does, the statement is so complex that it may be impossible to prove. Therefore, the validity of Equation (Equation 19) is commonly postulated together with the assumption that there is an invariant probability distribution μ on M, such that a measurement yields Equation (Equation 4) for μ -almost every x∈M, and for every phase function O. What is obvious is that dense exploration of the hugely dimensional phase space cannot be the reasons why experience taught us that Equation (Equation 4) can be used, at least in the case of equilibrium systems, with the classical ensembles as probability distributions.

This fact can be explained without invoking the ergodic properties E1–E4. Given a single variable O, its time average may appear like a full phase space average, even if its phase space trajectory $\Phi^{t}\Gamma$ explores a limited region of M. It suffices that the range of values of O be visited with proper frequencies; then the result is the same. Does this ever happen? Khinchin proved that this does indeed happen for the relevant observables of rarefied gases [29]. He noted that the subtleties of microscopic dynamics count little compared to the following:**(a)** macroscopic systems are made of very many particles: N⋙1;**(b)** only several and special phase functions are physically relevant;**(c)** it does not matter if ensemble averages disagree with time averages on limited sets of trajectories.

For rarefied gases, the relevant phase functions are sums of molecular contributions, $O\left(\Gamma \right)=\sum_{n=1}^{N}o\left(q_{n},p_{n}\right)$, where $\left(q_{n},p_{n}\right)$ is the vector of configurations and momentum of the *n*-th particle. These functions are appropriate for the pressure, temperature and density of rarefied gases, whose energy can also be expressed as $H=\sum_{n=1}^{N}h\left(q_{n},p_{n}\right)$, because interactions among particles are energetically negligible. (Interactions are, however, essential for the condition of LTE, hence for the existence of thermodynamic properties, to be established.) Then, in the microcanonical case, which here means a uniform probability distribution in M, Khinchin proved the validity of the following relation:(20)$$\mathrm{Prob}\left(\frac{|\bar{O}-E_{\mathrm{mc}}\left(O\right)|}{|E_{\mathrm{mc}}\left(O\right)|}\geq K_{1}N^{-1/4}\right)\leq K_{2}N^{-1/4}\;,$$
where $K_{1}$ and $K_{2}$ are positive constants, $\bar{O}$ is the time average of O, and $E_{\mathrm{mc}}\left(O\right)$ its phase space average in the microcanonical ensemble. In other words, the probability (in the microcanonical sense) that time averages differ by a small amount from the phase space averages is small if *N* is large: the larger *N* the smaller the probability of even smaller differences.

In this framework, the physically relevant ergodicity follows by and large from the N⋙1 condition, combined with the validity of the law of large numbers. That makes the range of sum variables of interest quite narrow, close to a single constant, when the single molecular contributions *o* can be considered independent, identically distributed, random variables. Then, exploration of the range of all the observables of interest does not take a long time. (See, e.g., Ref. [30] for an interesting analysis of the observables that relax to an equilibrium state, while others do not relax, because the underlying Toda dynamics are integrable, hence, not ergodic.) The details of the microscopic dynamics become irrelevant, while ensembles, i.e., probabilities in phase space, turn considerably useful. Even though probability, per se, is an “immaterial” and abstract mathematical notion, it becomes quite useful in the description of equilibrium states, under the above conditions. With the derivation of the Boltzmann equation a further step is taken forward, allowing time evolution.

## 5. Boltzmann Equation from the Probabilistic BBGKY Hierarchy

Let us specify the above general treatment of probabilities in phase space to the case of *N* identical spherical hard particles of mass 1. These particles are subjected to no external forces, and only interact elastically when they collide. We may think of their interaction potential as being zero when the particle centers are at a distance larger than their diameter, and infinitely high when this distance equals their diameter, so that particles are neither deformed, nor do they penetrate each other, when they collide. Denote a probability density in the phase space M by
(21)$$f_{N}^{\left(N\right)}\left(\Gamma ;t\right)=f_{N}^{\left(N\right)}\left(q_{1},p_{1};\ldots ;q_{N},p_{N};t\right)$$
with notation indicating the joint probability of *N* particles out of *N*. In absence of external forces, there are no accelerations between collisions; hence, ${\dot{p}}_{i}=0$ for every particle, and $\dot{\Gamma }=\left({\dot{q}}_{1},0;\ldots ;{\dot{q}}_{N},0\right)$. The Liouville equation can be written as
(22)$$\frac{\partial f_{N}^{\left(N\right)}}{\partial t}+\left({\mathrm{div}}_{\Gamma }\dot{\Gamma }+\dot{\Gamma }\cdot \nabla_{\Gamma }\right)f_{N}^{\left(N\right)}=\frac{\partial f_{N}^{\left(N\right)}}{\partial t}+\sum_{i=1}^{N}p_{i}\cdot \nabla_{q_{i}}\;f_{N}^{\left(N\right)}=0$$
where we have used the fact that the dynamics is Hamiltonian; hence, the first term in brackets is null. If we introduce the *s*-particle distribution function
(23)$$f_{N}^{\left(s\right)}\left(q_{1},p_{1};\ldots ;q_{s},p_{s};t\right)=\int dq_{s+1}dp_{s+1}\ldots dq_{N}dp_{N}\;f_{N}^{\left(N\right)}\left(q_{1},p_{1};\ldots ;q_{N},p_{N};t\right)$$
and we integrate the Liouville equation over the variables $q_{s+1},p_{s+1}\ldots q_{N},p_{N}$, we obtain [31]:(24)$$\frac{\partial f_{N}^{\left(s\right)}}{\partial t}+\sum_{i=1}^{s}p_{i}\cdot \nabla_{q_{i}}\;f_{N}^{\left(N\right)}=F^{\left(s\right)}\left(f_{N}^{\left(s+1\right)},f_{N}^{{\left(s+1\right)}^{\prime }}\right)$$
where $F^{\left(s\right)}$ is a function of the joint (s+1) particles distribution before collision $f_{N}^{(s+1)\prime }$, and of the corresponding distribution after collision $f_{N}^{\left(s+1\right)}$. Therefore, Equation (Equation 24) is not closed: computing $f_{N}^{\left(s\right)}$ requires knowledge of $f_{N}^{\left(s+1\right)}$. The set of distributions $f_{N}^{\left(s\right)}$ and their evolution equations, Equation (Equation 24), constitute the *BBGKYhierarchy*. In particular, taking s=1 and making $F^{\left(1\right)}$ explicit by solving the elastic collision dynamics, one obtains
(25)$$\frac{\partial f_{N}^{\left(1\right)}}{\partial t}+p\cdot \nabla_{q}\;f_{N}^{\left(1\right)}=\left(N-1\right)\sigma^{2}\int_{R}\int_{S_{-}}\left[f_{N}^{{\left(2\right)}^{\prime }}-f_{N}^{\left(2\right)}\right]\left|(p-p_{*})\cdot n\right|dp_{*}dn$$
for the one-particle probability distribution function, obtained integrating out the coordinates and momenta of all particles but one. Here, σ is the diameter of one particle; hence, $\sigma^{2}$ is the collision cross section; n is the unit vector in the direction going from the center of the test particle of position **q** and momentum **p**, labelled by 1, to that of another particle of momentum $p_{*}$; the range of integration $S_{-}$ is the hemisphere within which $(p-p_{*})\cdot n<0$, i.e., particles directed toward each other before collision; and (N−1) is the number of particles with which the test particle, can collide, assuming they all have the same probability of doing it, having integrated over all their momenta $p_{*}$. Moreover, having denoted by $f_{N}^{\left(2\right)}$ the two-particle distribution function, whose expression immediately before and after the collision respectively takes the form
(26)$$f_{N}^{{\left(2\right)}^{\prime }}=f_{N}^{\left(2\right)}\left(q,p^{\prime };q+\sigma n,p_{*}^{\prime };t\right)\;\mathrm{and}\;f_{N}^{\left(2\right)}=f_{N}^{\left(2\right)}\left(q,p;q+\sigma n,p_{*};t\right)$$
where $q_{*}=q+\sigma n$ gives the position of the second particle at the moment of collision, and
(27)$$p^{\prime }=p-\left[n\cdot (p-p_{*})\right]n\;\mathrm{and}\;p_{*}^{\prime }=p_{*}+\left[n\cdot (p-p_{*})\right]n$$
where $p^{\prime }$ and $p_{*}^{\prime }$ are, respectively, the momenta of the test particle and of the other colliding particle before collision, while p and $p_{*}$ denote the corresponding momenta after collision.

To solve this equation, one needs a closure assumption on $f_{N}^{\left(2\right)}$. The equation
(28)$$f_{N}^{\left(2\right)}\left(q,p;q_{*},p_{*};t\right)=f_{N}^{\left(1\right)}\left(q,p;t\right)f_{N}^{\left(1\right)}\left(q_{*},p_{*};t\right)$$
is known in the kinetic theory of gases as the *stosszahlansatz*, or hypothesis of molecular chaos. It amounts to assuming that when two particles collide, they are independent. As the dynamics of particles is deterministic, this statement may only have a statistical meaning, justified by some kind of dynamical *randomness*, that can be legitimately called *chaos*. (In the theory of dynamical systems, the term chaos is often used to indicate systems that have at least one positive Lyapunov exponent. This notion was not available to Boltzmann and, indeed, he did not need this kind of chaos.) Then, writing *f* in place of $f_{N}^{\left(1\right)}$ to simplify the notation, and recalling that v=p because we assumed unit masses, Equation (Equation 25) takes the form
(29)$$\frac{\partial f}{\partial t}+v\cdot \nabla_{q}\;f=\left(N-1\right)\sigma^{2}\int_{R}\int_{S_{-}}\left[f^{\prime }f_{*}^{\prime }-f\;f_{*}\right]\left|(v-v_{*})\cdot n\right|dv_{*}dn$$
where primes mean before collision and ∗ second particle, and small particles, small σ, imply $q_{*}\approx q$. This is the celebrated *Boltzmann equation*. Given its probabilistic derivation, which begins with the distribution $f_{N}^{\left(N\right)}$ in phase space, f(q,p,t)dqdp represents the probability that particle 1 be found in the volume element dq around q, with momentum within the set dp around p, at time *t*. The classical interpretation of probability we have so far adopted, then states that, given a very large collection of independent macroscopically equal systems, each made of *N* hard spheres, the fraction of them having particle 1 within dqdp equals fdqdp. If one assumes, in addition, that particles are indistinguishable, a simple combinatorial factor turns fdqdp into the fraction of ensemble members having one particle, whichever it is, in dqdp at time *t*. It remains to connect this probability to material properties, like the density of the gas in dqdp.

## 6. Boltzmann Equation from Mass Balance in Real Space

The probability distribution *f* appearing in Equation (Equation 29) is obtained projecting the one for *N* particles defined on the corresponding 2dN dimensional phase space, down to the 2d dimensional one-particle space. Given a single system of interest and the ensemble interpretation of the probability in phase space, the condition $f_{N}^{\left(1\right)}\left(q,p;t\right)>0$ does not imply that particle 1 actually lies within the volume dqdp, since only a fraction of the systems of the ensemble has particle 1 in dqdp: particle 1 of the complementary fraction of systems does not lie in dqdp. (If particles are indistinguishable and the 1-particle probability distribution has been correspondingly normalized, the same can be stated for any single particle of the systems of the ensemble.) In particular, dqdp could be empty because the BBGKY hierarchy can be developed for any N≥1, and a few particles only occupy a limited region of space. Therefore, one could conclude that even the Boltzmann equation is *immaterial*, and there is no compelling reason to take it as a description of a given material system.

However, a physical derivation of the Boltzmann equation is also available. It does not start from a probability distribution on a phase space M, which is a continuum of geometric points, each of which represents a whole *N*-particles system. This derivation refers to a single system made of a very large number *N* of small, but finite size, particles of mass 1, with positions and velocities in a given volume ΔqΔp. The particles in such a volume constitute a discrete rather than a continuous set, and can be represented by a continuous density of mass only when many of them populate ΔqΔp. Therefore, given the finite size of the particles, such a volume has to be small, to appear like a point on the macroscopic scale. However, unlike the volumes dqdp of the previous section, which can be arbitrarily small, it cannot be infinitesimal. Indeed, it has to be much larger than the volume of a single particle, so that many of them can be lodged in it [32,33]. This highlights another essential difference between probability and mass densities. Let us denote by $C_{ij}$ of size ΔqΔp one cell centered around a discrete set of lattice points $\left(q_{i},p_{j}\right)$. The mass density within $C_{ij}$ is defined as the mass of one particle times the number density $\rho (q_{i},p_{j};t)$, the number of particles per unit volume around $\left(q_{i},p_{j}\right)$. Letting $n_{ij}$ be the number of particles in cell $C_{ij}$, at time *t*, one may thus write
(30)$$n_{ij}\left(t\right)=\rho \left(q_{i},p_{j};t\right)\Delta q\Delta p\;;\;\sum_{i,j}n_{ij}\left(t\right)=N$$
and may try to approximate this set of discrete mass densities using a continuous function ρ, so that
(31)$$N=\int_{V_{\mu }}\rho \left(q,p;t\right)dqdp$$
where $V_{\mu }$ is the set of positions and momenta available to particles of the system. This makes no sense unless $n_{ij}$ is very large in each cell $C_{ij}$; hence, such cells must be relatively large. Furthermore, particles have to collide: their motion has to be random, so that a uniform distribution can be rapidly achieved within each $C_{ij}$. Large $n_{ij}$, together with dynamical randomness, makes the particles entering or leaving a cell just a negligible number, compared to those staying inside. At the same time the cells must be sufficiently small compared to the observation scale, that they appear like a point. A wide separation of scales, possible only if *N* is very large, is required.

Let us now follow the motion of one set of particles lying in one cell ΔqΔp at time *t*, as they stroll in $V_{\mu }$. Let us assume that ρ accurately describes the mass density as a continuum. In the absence of external forces, collisions are the only mechanism that allows velocities to change. If, in a time dt, the particles in ΔqΔp undergo no collisions, the position q of one of them turns q+vdt, while its momentum p is unchanged, and all the particles in the cell ΔqΔp, centered about (q,p), will be found in the cell $\Delta \hat{q}\Delta \hat{p}$ centered about (q+vdt,p). This is achieved by evolving in time all points of the original cell with the same dynamics of the particles in it. Therefore, in the absence of collisions, one has
(32)$$\rho \left(q+vdt,p,t+dt\right)\Delta \hat{q}\Delta \hat{p}=\rho \left(q,p,t\right)\Delta q\Delta p$$
As q and p are canonically conjugate variables, and between two consecutive collisions each particle can be considered by itself as a whole Hamiltonian system, we obtain $\Delta \hat{q}\Delta \hat{p}=\Delta q\Delta p$, and
(33)ρ(q+vdt,p,t+dt)=ρ(q,p,t)
On the other hand, if particles collide, one additional term arises:(34)$$\rho \left(q+vdt,p,t+dt\right)=\rho \left(q,p,t\right)+{\left(\frac{d\rho }{dt}\right)}_{coll}dt$$
which defines the collision term. Finally, expanding the left-hand side of Equation (Equation 34) to first order in dt, one obtains
(35)$$\rho \left(q,p,t\right)+\frac{\partial \rho }{\partial t}dt+v\cdot \nabla_{q}\;\rho dt=\rho \left(q,p,t\right)+{\left(\frac{d\rho }{dt}\right)}_{coll}dt$$
because ρ depends on *t* both explicitly and implicitly through the position coordinates. Dividing by dt yields:(36)$$\frac{\partial \rho }{\partial t}+v\cdot \nabla_{q}\rho ={\left(\frac{d\rho }{dt}\right)}_{coll}$$
where $v\cdot \nabla_{q}\rho$ represents streaming in position space. There is no streaming in momentum space, because no field acts on the particles, but particles may enter or exit Δp thanks to collisions.

Relation (Equation 36) expresses the conservation of mass, and does not require any interpretation of a probabilistic nature: it is based on the objective counting of particles (or measurement of mass density) and on their deterministic Hamiltonian dynamics. Assuming that instantaneous collisions of three or more particles are negligible in a gas of hard spheres, the collision term should be expressed as a function of the number of pairs of particles located inside Δq that may either enter (gain term) or leave (loss term) Δp because of collisions. Let $\rho^{\left(2\right)}$ be the density of such pairs. As the initial condition of the *N* particles is not known, going beyond Equation (Equation 36) requires some assumption on the collision term, analogously to Equation (Equation 25). One may then rely on the large value of *N*, on the smallness of the particles, and on the disorder produced in configuration and velocity space by the collisions, which make sensible the factorization of $\rho^{\left(2\right)}$:(37)$$\rho^{\left(2\right)}\left(q,p;q+\sigma n,p_{*};t\right)=\rho^{\left(1\right)}\left(q,p;t\right)\rho^{\left(1\right)}\left(q+\sigma n,p_{*};t\right)$$
That, indeed, means that each of the $n_{\Delta q}$ particles in an elementary volume Δq may collide with any of the remaining $n_{\Delta q}-1\approx n_{\Delta q}$ either for a gain or a loss of particles in the elementary volume Δp. This is a rarefied gas condition, because it means that no particle can hide behind another one. Then, integrating over all possible velocities yields an expression like Equation (Equation 29), which makes sense only if particles are many and small. The equation for the mass distribution turns identical to Equation (Equation 29), introducing $\rho =\rho^{\left(1\right)}/N$. Therefore, the Boltzmann equation is obtained again. Its solution ρ enjoys the same properties that *f* enjoys.

At the same time, the conceptual difference between *f* of Equation (Equation 25) and ρ of Equation (Equation 36) could not be greater. In the first case one considers the fraction of identical abstract systems that enjoy certain properties out of the total of an ideal hypothetical continuum of identical systems; in the second case one considers the number of particles of a single concrete, experimentally observable system. Probabilistically, all calculations can be performed for a large collection of identical systems made of any number N≥1 of particles; physically, a single systems is considered and *N* must be huge. At this level of description, there is no stringent reason for one quite sophisticated abstract object like *f* to behave similarly to a material tangible object, such as ρ. Nevertheless, this is precisely what happens if probability is required to evolve like a fluid made of geometric points obeying the equations of motion of the system that each of them separately represents, and the *stosszahlansatz* holds. That may well be mathematically too hard to prove within a general Hamiltonian framework, also because it is not verified in a series of systems, hard to fully classify, cf. integrable systems. However, experience has demonstrated that it does hold in very many situations; situations in which we may legitimately state that *probability* turns *material*: probability can be formally identified with mass.

What is the cost of such an identification? When can the stosszahlansatz be justified? In the first place, one would think that large *N* makes particles independent before colliding, because they may have hardly met before, but this is not enough. Particles are not independent after having collided with each other. Moreover, the factorization condition
(38)$$f_{N}^{\left(s\right)}\left(q_{1},p_{1},\ldots ,q_{s},p_{s};t\right)=\prod_{j=1}^{s}f_{N}^{\left(1\right)}\left(q_{j},p_{j};t\right)$$
is hard to obtain because particles do not overlap, and *s* of them occupy a volume of order $O(s\sigma^{3})$. Thus, large *s* increases the region to be excluded from the free motion of particles, resulting in correlations among them that may persist over time. This suggests that s=2 is the best candidate for the postulated independence. Also, large *N* helps in making rare the encounters between the same pair of particles only if the density is small; otherwise packed particles may collide forever with the same small group of companions. Then, σ should be small. On the other hand, collisions of hard spheres are defocussing, and have a randomizing effect on positions and velocities, which is beneficial for the decay of correlations. Therefore, *N* and σ should be tuned so that the total cross section is not negligible. Grad identified the scaling regime in which all the above is realized, now known as Boltzmann–Grad limit [34]. It consists in keeping $N\sigma^{2}$ positive and finite, while *N* grows:(39)$$N\sigma^{2}=\mathrm{constant}>0\;\mathrm{which}\;\mathrm{implies}\;\sigma^{2}∼\frac{1}{N}\;,\;\;N\sigma^{3}∼N^{-\frac{1}{2}}$$
In this limit, the volume occupied by particles is negligible, while the total cross section is finite. More precisely, let λ denote the mean free path and τ the mean free time as, respectively, the typical microscopic length scale of the system, which is the distance traveled on average by a particle between collisions, and the corresponding mean time. The first is defined by the the ratio between the volume and the total cross section of the particles, the second by the ratio between this distance and the typical speed. Then, denoting by |Ω| the available volume, and letting $\beta =1/k_{B}T$, so that the typical speed for particles of mass 1 is $\sqrt{2/\beta }$, we have
(40)$$\lambda =\frac{\left|\Omega \right|}{N\pi \sigma^{2}}\;,\;\tau =\sqrt{\frac{\beta }{2}}\frac{\left|\Omega \right|}{N\pi \sigma^{2}}$$
which remain finite in the Boltzmann–Grad limit. This implies a net effect produced by particle collisions, which is necessary to randomize their motion. Together with the vanishing of the excluded volume, the limit of Equation (Equation 40) is also necessary to make correlations rapidly decay, when *N* is large. Physically, this picture corresponds to a rarefied gas, in which particles collide, but the time average of their interaction energy is negligible compared to the time average of their kinetic energy. Whether the assumptions apply or not, only experience can tell. An important result implied by the Boltzmann–Grad limit is that $f_{N}^{\left(2\right)}$ remains factorized in time, if the solution of the *s*-th equation of the BBGKY hierarchy, $f_{N}^{\left(s\right)}$, exists is unique, and starts factorized as *s* factors $f_{N}^{\left(1\right)}$ for all *s* [31].

Arguably, the most relevant conceptual result stemming from the Boltzmann equation is the *H*-theorem. It can be concisely illustrated as follows: let the *H*-functional be defined by
(41)$$H\left(f\right)=\int_{V\times R^{3}}f\log\left(f\right)\;dpdq$$
where $V\subset R^{3}$ is the volume in which a gas is contained. For *f* a solution of the Boltzmann equation within the volume *V*, the *H*-theorem states
(42)$$\frac{d}{dt}H\left(f\right)\leq 0$$
where the equality holds if and only if *f* is constant for q∈V and the **p** are distributed as a Maxwellian probability distribution.

A point that may at times be overlooked, is that the solution of the Boltzmann equation requires a specification of initial and boundary conditions, like any other integro-differential equation. The form of the boundary of *V* is crucial, particularly for the validity of the *H*-theorem and, in general, for the overall solution. For instance, reflecting or periodic boundaries are commonly used, and the boundary is typically assumed to be piece-wise smooth, although works on irregular boundaries have been performed, cf. [35]. Moreover, *V* must be compact for physical properties such as pressure to have a definite equilibrium positive value.

The *H*-theorem shows that the Boltzmann equation exhibits irreversible behaviour, which appears in contrast to the time-reversal invariant laws of classical mechanics governing the motion of particles. In reality, various approximations and hypotheses, which are extraneous to the dynamics of an *N* particle system, have been introduced to pass from the evolution of probabilities in phase space to the Boltzmann equation. The *N* particle system has been approximated by a system consisting of infinitely many particles, verifying the molecular chaos hypothesis, through the Boltzmann–Grad limit, thus breaking time reversibility. The interesting fact is that such approximations and hypotheses perfectly represent the physics of a rarefied gas.

We conclude this section by noting that there exists quantum mechanical and relativistic generalizations of the Boltzmann equation, cf., e.g., Refs. [7,36]. These generalizations have proven successful in an exceedingly wide range of applications, including transport of electrons in solid conductors and in astrophysical plasmas. The most direct extensions are the classical ones, specifically concerning particles that are accelerated by external fields, such as electric fields. In this case, streaming in velocity space is introduced in addition to the streaming in position space. Denoting by F the external force and by *m* the mass of the particles, its effect on v appears as the acceleration F/m. This acceleration smoothly guides a particle inside or outside an element of velocity space, analogous to how velocity guides a particle inside or outside an element of position space. Consequently, the Boltzmann equation may be re-written as
(43)$$\frac{\partial \rho }{\partial t}+v\cdot \nabla_{q}\rho +\frac{F}{m}\cdot \nabla_{v}\rho ={\left(\frac{d\rho }{dt}\right)}_{coll}$$
In sufficiently hot plasmas made of charged particles of charge *c*, collisions can be neglected. Thus, denoting the electric and magnetic fields acting on these particles as E and B, respectively, Equation (Equation 43) can be re-written as
(44)$$\frac{\partial \rho }{\partial t}+v\cdot \nabla_{q}\rho +\frac{c}{m}\left(E+v\times B\right)\cdot \nabla_{v}\rho =0\;,$$
which is known as the Vlasov equation. In reality, variations of the Boltzmann equation are numerous. The kinetic theory of wave turbulence [37] finds applications from quantum to astrophysical phenomena, including anomalous transport. There are Boltzmann equations for thermostatted rather than Hamiltonian dynamics, and for a wide spectrum of variously interacting systems, such as the components of blood, or the individuals in a social network etc. It is impossible to list all of them. Of course, there is a difference between these variations and the original Boltzmann equation: the interaction rules are often rather loosely known, which makes the fundamental hypothesis of molecular chaos hard to justify. Nevertheless, these variations provide valuable insights into non-standard and even non-physical phenomena. The interested readers may easily find the relevant literature, which is not within our scope.

## 7. Concluding Remarks

In this paper we have reviewed some of the reasons that, in principle, could make probability an abstract physically inapplicable mathematical tool, while it is clear that it is arguably the most effective tool physicists can use. The point is that such effectiveness does not arise for free; it is the result of informed, repeatedly tested, and verified choices, as revealed by the derivation of the Boltzmann equation. Uncritical use of probabilities can easily lead to errors. Keeping in mind the teachings of statistical mechanics, which have unfolded since Boltzmann’s glorious work of 180 years ago, is thus beneficial for future developments, which will venture in largely uncharted territory, often at variance with our established knowledge of macroscopic objects. Indeed, the microscopic fluctuations are not observable in the behaviour of macroscopic objects in thermodynamic conditions, although present day technology allows us to measure them in microscopic observations [38,39]. On the contrary, fluctuations may even dominate the behaviour of small systems, and require suitable approaches, whose results must be properly interpreted, especially if one insists on using the thermodynamic language. For instance, when dealing with situations in which there is no dissipation, one may find that the derivations of thermodynamic relations are only formal, and that the relevant quantities only have a statistical meaning. However, even more problematic may be the extension to nonequilibrium (dissipative or non-dissipative) states, where even the formal thermodynamic analogies may fail [40]. At present, one must proceed by examples, as a comprehensive theory of small and nonequilibrium systems is still missing, although notable works are being performed, see, e.g., Ref. [41] for one review.

## Data Availability

Data are contained within the article.

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
