# Peer review of "Probability Turns Material: The Boltzmann Equation"

_entropy, 2024, doi:10.3390/e26020171_

Round 1

Reviewer 1 Report

Comments and Suggestions for Authors

The authors analyze how the concept of probability has to be interpreted in the context of the Boltzmann equation. In particular, they review two derivations of it: one in which the probability is interpreted as an average over different realization (it can be considered to be "immaterial"), and other in which probability is defined in terms of a single realization. In the last case, the probability is interpreted as the mean value of the number of particles over large enough cells in the phase space (so, the probability turns "material" within this interpretation).

I think that the paper offers a good review of these (often overlooked) concepts and that frequently lead to misconceptions in the interpretation of the results. The paper is well written and the concepts and results are presented in a clear way. Then, I can  recommend the paper for publication.

I only have a comment/suggestion:

1) The Boltzmann equation has to be solve in certain volume in space with the appropriate boundary conditions.

2) In the Boltzmann's formulation of the equation, the possible existence of an external field is considered.

Why these concepts are not mentioned in the paper? In my opinion, they are essential in the context of the Boltzmann equation and do not affect to the interpretation of the concept of probability. I suggest the authors to write some words about it.

Minor comments:

- After eq. (17): "... from the observation that a they take ..."

- After eq. (26): I think the definition of $q*$ is not necessary as it is explicitly written in the argument of $f_N^{(2)}$.

Author Response

Thank you sincerely for dedicating time to review our manuscript. We genuinely appreciate your thoughtful insights and valuable suggestions. We have diligently incorporated your recommendations into our work, and we believe they have significantly enhanced the overall quality of the manuscript.
We have addressed your comment by adding some details in Section 6, which provides further explanation and context. Please refer to the updated manuscript for details.

Reviewer 2 Report

Comments and Suggestions for Authors

In this work the authors revise the significance of the Boltzmann equation (BE) to understand the dynamics of many-particle systems and its link with deep  mathematical and physical problems. The authors highlighted the fundamental role of probability theory in physical systems together with the paradigm change brought by the BE in connection with the propagation of chaos when the number of particles becomes large.  

The article is very well written and sound and for these reasons I can recommend its acceptance.

Anyway I would like to suggest a brief discussion on two missing aspects:

1) the study of the trends to equilibrium of the Boltzmann equation, also in connection with the so-called Cercignani conjecture. In this direction, the study of entropy dissipation had a very significant impact in the mathematical community in connection with the fluid limits of the Boltzmann equation. 

2) the connection of Boltzmann-type equations with new fields in socio-economic and life sciences where universal physical laws are not available and data typically exists in the macroscopic scale. 

Author Response

Thank you sincerely for dedicating time to review our manuscript. We genuinely appreciate your thoughtful insights and valuable suggestions. We have diligently incorporated your recommendations into our work, and we believe they have significantly enhanced the overall quality of the manuscript.
We have addressed your comment by adding some details in the introduction, which provides further explanation and context. Please refer to the updated manuscript and the attachment for details.
